# Influenza vaccination hesitancy in five countries of South America. Confidence, complacency and convenience as determinants of immunization rates

**Miguel Ángel González-Block**[1,2,3]*, **Emilio Gutiérrez-Calderón**[4], **Blanca Estela Pelcastre-Villafuerte**[2], **Juan Arroyo-Laguna**[5], **Yamila Comes**[6], **Pedro Crocco**[7], **Andréa Fachel-Leal**[8], **Laura Noboa**[9], **Daniela Riva-Knauth**[8], **Berenice Rodríguez-Zea**[3], **Mónica Ruoti**[3], **Elsa Sarti**[10], **Esteban Puentes-Rosas**[11]

1 Universidad Anáhuac, Mexico City, Mexico, 2 Instituto Nacional de Salud Pública, Cuernavaca, Mexico, 3 Evisys Consulting, Mexico City, México, 4 Universidad Nacional Autónoma de México, Mexico City, Mexico, 5 Pontificia Universidad Católica del Perú, Lima, Peru, 6 Maestría en Sistemas de Salud y Seguridad Social, Universidad ISALUD, Buenos Aires, Argentina, 7 Escuela de Salud Pública de Chile, Santiago, Chile, 8 Universidade Federal do Rio Grande do Sul, Porto Alegre, Brazil, 9 Facultad de Ciencias Sociales, Universidad de la República, Montevideo, Uruguay, 10 Sanofi Pasteur LATAM, Mexico, 11 Sanofi Pasteur LATAM, Panama

* miguel.gonzalezblock@gmail.com

**Data Availability Statement:** The data underlying the results presented in the study are available

## Abstract

### Introduction

Influenza morbidity and mortality are significant in the countries of South America, yet influenza vaccination is as low as 56.7% among pregnant women, reaching 76.7% of adults with chronic diseases. This article measures the relative values for the vaccination hesitancy indicators of confidence, complacency and convenience by risk-groups in urban areas of five countries of South America with contrasting vaccination rates, analyzing their association with sociodemographic variables and self-reported immunization status.

### Methods

An exit survey was applied to 640 individuals per country in Brazil, Chile, Paraguay, Peru and Uruguay, distributed equally across risk groups of older adults, adults with risk factors, children ≤6 and pregnant women. Indicators were constructed for vaccine confidence, complacency and convenience. Analysis of variance and multiple logistic analysis was undertaken.

### Results

Adults with risk factors are somewhat more confident of the influenza vaccine yet also more complacent. Convenience is higher for mothers of minors. Children and older adults report higher levels of vaccination. The 3Cs are more different across countries than across risk groups, with values for Chile higher for confidence and those for Uruguay the lowest.

from Harvard Dataverse under the link: https://dataverse.harvard.edu/dataset.xhtml?persistentId=doi:10.7910/DVN/9O6LPP.

**Funding:** Sanofi Pasteur provided support for this study in the form of a grant to Evisys Consulting and in the form of salaries for ES and EPR. Evysis Consulting provided support in the form of salaries to MAGB, EGC, BEP, JA, YC, PC, AFL, LN, DRK, BRZ and MR. BRZ and MR participated in the research as paid, pro-tempore employees of Evisys. EGC, BEP, JA, YC, PC, AFL, LN, DRK participated as Evisys consultants. The specific roles of these authors are articulated in the 'author contributions' section. The grant provided by Sanofi Pasteur to Evisys was used to pay consulting fees and salaries and to pay for the data acquisition. Both Sanofi Pasteur and Evisys participated in study design and manuscript review, but had no further role in data collection and analysis, decision to publish, or preparation of the manuscript.

**Competing interests:** The authors have read the journal's policy and the authors of this manuscript have the following competing interests: ES and EPR are paid employees of Sanofi Pasteur and MAGB, BRZ and MR are paid employees of Evisys Consulting. BRZ and MR participated in the research as paid, pro-tempore employees of Evisys. EGC, BEP, JA, YC, PC, AFL, LN, DRK participated as Evisys consultants. Both Sanofi Pasteur and Evisys participated in study design and manuscript review. This does not alter our adherence to PLOS ONE policies on sharing data and materials. There are no patents, products in development or marketed products associated with this research to declare.

Complacency is lower in Brazil and higher in Uruguay. Results suggest that confidence and complacency affect vaccination rates across risk groups and countries.

## Conclusions

Influenza vaccine confidence, complacency and convenience have to be bolstered to improve effective coverage across all risk groups in the urban areas of the countries studied. The role played by country contextual and national vaccination programs has to be further researched in relation to effective coverage of influenza vaccine.

## Introduction

Influenza is a respiratory viral disease that is responsible for high mortality and morbidity rates worldwide [1, 2]. The disease can be particularly severe in children younger than 6 years, pregnant women, older adults, and adults with risk factors [3]. Influenza cases are estimated at 1 billion per year worldwide, of which 3 to 5 million are severe and lead to 290,000–650,000 deaths [1]. Vaccination is one of the most important measures to prevent influenza infection. Influenza vaccine has been reported with up to 90% of effectiveness in the case of healthy adults, while in older adults, effectiveness can be of 60% while reducing mortality by up to by 80% [4]. Some years, nevertheless, influenza vaccine can have a low performance due to a mis-match with circulating strains [5].

Most Latin American countries have a seasonal influenza vaccination policy in place and the vaccine is recommended and publicly financed for children aged under 6 years, people with chronic comorbidity, adults aged over 60 years, pregnant women within 20 weeks of gestation, or women in the postpartum period [6]. WHO recommends the attainment of 90% coverage for all universal vaccinations for 2020 [7]. Yet among 10 countries of South America including Brazil and those in the Andean Region and the South Cone, reported coverage for influenza vaccination in 2018 (or the most recent year) was 61.6% of adults 60 years and older, 58.4% of children under six years of age, 56.7% of pregnant women and 76.7% of adults with chronic conditions [8]. Vaccination coverage across all risk groups in Argentina, Bolivia, Brazil, Chile, Colombia and Ecuador perform above the regional average, with Paraguay, Peru, Uruguay and Venezuela performing below average.

The decision-making process followed by the population to get vaccinated is immersed in a specific social context of beliefs and perceptions as well as considerations of the availability of the vaccine and its costs [9]. The World Health Organization's Strategic Advisory Group of Experts proposed the concept of vaccine hesitancy defined as the delay in the acceptance or to the rejection of vaccines despite their availability within vaccination services [10]. Vaccine hesitancy is the result of a complex interrelation of behavioral and societal factors whose intervention requires an integral approach. Different conceptual models have been proposed to address the complexity, applicability, and potential usefulness of vaccine hesitancy indicators, as well as for the design of surveys and interventions that can be applied locally and globally. The "Three Cs" model of vaccine confidence, complacency, and convenience is considered as one of the most useful given that it is intuitive and easy to understand and apply [11]. Confidence is the degree of trust in the effectiveness and safety of the vaccine, in the system that delivers the vaccines, and in the motivations of those who make the decisions to achieve effective access to the vaccines. Lack of confidence is caused by strong negative attitudes towards vaccination, which can be influenced by misinformation about vaccination risks, by affiliation

to anti-vaccine groups or through legitimate concerns regarding vaccine safety and efficacy. Complacency occurs when the risk of diseases preventable by vaccination is perceived as low, and vaccination is not considered a necessary or the chief preventive measure. Complacency is influenced by the relative priorities assigned to health as against other life responsibilities, including the success of immunization programs and their impact in lowering disease preva- lence and mortality and self-efficacy–the self-perceived or real ability of an individual to take action to vaccinate. Convenience refers to the influence on the decision to get vaccinated of vaccine availability, affordability, willingness to pay, ability to understand and accept vaccine- related information (language, culture and health literacy), health service quality (real or per- ceived) and the degree to which vaccination services are delivered at a time and place and in a cultural context that is convenient and comfortable [12].

Influenza vaccine hesitancy can be of greater significance for coverage in comparison to other vaccines given its seasonal character, annual variability in effectiveness and often low perceived levels of protection, abundance of influenza-specific myths, and its recommendation for specific risk-groups [13]. The 3C model of vaccine hesitancy has been applied to compare across influenza and other vaccines as well as to explore and compare the role its components play for influenza vaccine uptake. A world-wide systematic review of influenza vaccination intention and behavior studies between 2005 and 2016 identified important barriers to vacci- nation against influenza in all risk groups. The most frequent reasons for hesitancy were due to low perceived risk of the disease, lack of trust in the authorities, and low perceived safety of the vaccine [14]. Crouse Quinn and colleagues developed quantitative indicators for each of the 3C components and found significant effects with influenza vaccine hesitancy in represen- tative samples of African American and native white populations in the United States, as well as different measures of hesitancy across vaccines for different diseases [15].

The role of confidence, complacency and convenience for increasing influenza vaccination coverage needs to be further researched in specific national and local contexts and across risk groups to identify the specific factors that can modified to reduce vaccine hesitancy and to increase coverage. Quantitative measurement of each of the 3C vaccine hesitancy components can facilitate the identification of context-sensitive factors associated to vaccination uptake. In this article we explore three research questions: how different are confidence, complacency and convenience across risk groups and countries? How are the sociodemographic characteris- tics of risk-group members related to each of the three components of hesitancy? ¿How sensi- tive are each of the components to self-reported vaccination uptake? To pursue these research questions, this paper proposed indicators for confidence, complacency and convenience and analyzes their association across risk groups and across selected countries of South America and in relation to socioeconomic characteristics and self-reported vaccination status.

## Methods

A cross-sectional quantitative study was designed to identify confidence, complacency and convenience as determinants of influenza vaccination hesitancy and their association to vacci- nation rates among low and middle-class residents of large cities in a sample of countries of South America. Five countries were selected based on judgement of their vaccination rates and aiming to include countries along the vaccination coverage continuum and to include coun- tries in Brazil, the Andean Region and the South Cone. Brazil and Chile were selected as high performers and Paraguay, Peru and Uruguay as low performers.

The "Three Cs" model was operationalized following Wheelock and collaborators protocol for integrating self-reported knowledge, attitudes and practices into vaccine confidence, com- placency and convenience indicators [10]. Pertinent and reliable vaccine hesitancy questions

were adapted for each of the 3C components from Larson et al, who developed a set of model survey vaccine hesitancy questions supported on a matrix of hesitancy determinants selected through a systematic review of peer-reviewed literature and on the expertise of the SAGE Working Group on Vaccine Hesitancy [10]. Some of these survey questions had been pilot-tested in various world regions including the Americas through a questionnaire in Spanish.

A questionnaire was designed for each risk group structured in 41 Core Closed and Likert Scale questions (available as S1 Questionnaire and S1 File). Most questions were the same across risk groups, except on the specifics of pregnancy and chronic disease self-report and the specifics of influenza and influenza vaccine risks for babies, for children, for older adults and for adults with risk factors. Questionnaires were adapted to each country's health care systems to ask about health insurance coverage.

Questionnaires were developed in Spanish and pilot tested in Peru. Questionnaires were then translated to Portuguese for application in Brazil and further pilot tested in each country to ensure they were properly understood in the urban settings where they were going to be applied. The questionnaire was applied to each risk group through opportunistic sampling in two public and two private ambulatory healthcare units for a total of eight units. The health centers were located in the following cities. In Brazil: Porto Alegre and Sao Paulo; in Chile: Santiago and Valparaiso; in Peru: Arequipa and Lima; in Paraguay: Asunción and Ciudad del Este, and in Uruguay: Montevideo and Salto. Primary care exit surveys were carried out in a sample of public and private health centers with a sample of adults aged 65 years and over, adults with risk factors, pregnant women, mothers of children aged <6 years. Exclusion criteria included major impairment due to illness. Adults with risk factors included participants 18 years and older who reported at least one underlying health condition, and which included at least one of the following: hypertension, gastritis, diabetes, cancer, chronic pulmonary diseases or depression. Research was undertaken from October 2018 to December 2019. The protocol was approved by research ethics committees in each of the countries studied (see detailed ethics committee information in Ethics Approval below).

The sample size was not calculated based on statistical error and confidence criteria because the selection process was not probabilistic and, therefore, was not intended to obtain estimates with associated levels of precision (error and confidence). The number of interviews was established based on optimization criteria seeking an adequate balance between the availability of resources and the robustness of the results. A minimum of 640 participants were surveyed in each country, 160 individuals for each risk group for a minimum grand total of 3,200 individuals.

Aggregated indicators for each of the 3C components were constructed through coding questions to each of the 3C indicators and averaging their component variables. Indicators for complacency subcomponents of prejudices, knowledge and risk perception were also constructed. Table 1 details the construction of each component and subcomponent. The vaccine confidence indicator was constructed based on three questions on vaccine safety and effectiveness, each based on a five-point Likert scale for up to 12 points. The vaccine complacency component and subcomponents were constructed based on nine questions for a total of up to 41 points among dichotomous and Likert scale responses as well as a list of up to fifteen influenza symptoms. One point was considered for each correct symptom mentioned. The vaccine convenience indicator was constructed adding values for five dichotomous responses related to vaccine recommendation and access with up to five points each, for a total of 5 points. Values within each of the 3Cs and in the case of Complacency within each sub-indicator were standardized in a scale of 0 to 10, with greater value for more confidence and convenience and less complacency. Within the complacency subcomponents, a greater value is allocated to less prejudice, more knowledge and more risk perception of influenza risks.

**Table 1. Construction of confidence, complacency and convenience indicators.**

| Indicator | Description | Questionnaire items* | Construction | Measurement scale (and points) |
|---|---|---|---|---|
| **Confidence in the vaccine** | Level of perception of the efficacy and safety of the vaccine | 24.1: "Vaccine efficacy level".<br>24.2: "Vaccine safety level".<br>25.1: "The vaccine is very effective" | The scale was additive according to ordinal answers obtained in questions. Resulting sum is rescaled to the interval (0, 10). | From low confidence (0 Pnts.) to high confidence (12 Pnts.) The more confidence, the less hesitation to vaccinate |
| **Complacency A. Influenza risk** | Level of perception of the risk of contracting influenza and its severity | 24.3: "Level of risk of contracting influenza.*<br>24.4: "Flu severity level". | Idem. | From low risk (0 Pnts.) to high risk (8 Pnts.) The greater the risk, the less hesitation to be vaccinated |
| **Complacency B. Knowledge of influenza and the vaccine** | Level of knowledge of influenza and its vaccine | 11: "You know what influenza is.*<br>12: "Main symptoms of influenza.*<br>13: "You know the vaccine exists"<br>25.5: "It is advisable to vaccinate against influenza every year."<br>25.6: "Only minors and the elderly should be vaccinated." (Calculated in an inverted sense to be consistent with the direction of the indicator) | The scale was additive for positive answers in each of the questions. Resulting sum is rescaled to the interval (0, 10). | From low knowledge (0 Pnts.) to high knowledge (25 Pnts.) The more knowledge, the less hesitation to vaccinate |
| **Complacency C. Vaccine prejudices** | Level of prejudices expressed about influenza vaccine | 25.2: "The vaccine has side effects."<br>25.4: "The vaccine causes reactions." | The scale was additive according to ordinal answers obtained in questions. Resulting sum is rescaled to the interval (0, 10). (The score complement is used to be consistent with the direction of the indicator) | From low prejudice (0 Pnts.) to high prejudice (8 Pnts.) A lower value, less prejudice and less hesitation to get vaccinated |
| **Convenience** | Level of convenience perceived in accessing the vaccine | 19. Who recommended you get vaccinated?<br>20. Do you know where to go to get a flu shot?<br>22. Is the vaccine available at the health facility where you go regularly?<br>23. How long does it take to get from your home to the health facility you go to regularly?<br>25.3 Is the flu vaccine difficult to obtain? | The scale was additive. Resulting sum is rescaled to the interval (0, 10). | From low convenience (0 Pnts.) to high convenience (5 Pnts.) The more convenience, the less hesitation to vaccinate |

* Reference is made to the questionnaire for Adults with Risk factors. Questionnaires for other risk groups specified question 25.6. Questionnaire is available as S1 Questionnaire.

One-way ANOVA pairwise multiple comparison tests were applied to analyze the significance of differences in means across risk groups and countries for the aggregate 3C indicator, the separate 3C components and subcomponents of complacency as well as of vaccination status (at least once in the life-course and in the last year). Analysis of variance was applied to assess the significance of the association between the 3C components and sociodemographic variables, risk groups and countries. Binary logistic multivariate regression analysis was applied to assess the association within each risk group between vaccine confidence, convenience and complacency and vaccination status. The processing and analysis of the information was carried out using the IBM-SPSS V.24 package.

## Ethics approvals

The following ethics committee reviewed the protocol within each of the study countries:
Brazil, Comissao Nacional de Ética em Pesquisa, 05215918.6.0000.5347.

Chile: Comité de Ética de Investigación en Seres, Universidad de Chile, Facultad de Medicina, 191–2018.

Paraguay: Comité de ética en Investigación, Laboratorio Central de Salud Pública, 106/2019.

Peru: Comité de Ética de Investigación Prisma, CE1651.18.

Uruguay: Comité de Ética en Investigación, Instituto Nacional de Salud Pública, 1580.

Mexico: Comité de Ética en Investigación, Instituto Nacional de Salud Pública, Proyecto CI: 1580.

## Results

Women constituted between 67.7 and 69.4% of respondents in the groups of older adults and adults with risk factors (Table 2). Mean age was 71.8 for older adults, 51 years for adults with risk factors, 27.8 for pregnant women and 30.5 for mothers of minor children. Education levels were higher among pregnant women and mothers of minor children than for adults with risk factors and older adults, with 8.1 and 9.3% of the former having up to primary education, as against 26.9 and 38.8% of the latter, respectively. Adults with risk factors reported as the most prevalent diseases hypertension (43%), diabetes (22%) and gastritis or gastric ulcer (14%). Across countries, average age was similar while Paraguay and Uruguay registered the largest percentages of respondents with up to primary education, while Uruguay had to second lowest level of respondents with higher education.

**Table 2. Sociodemographic characteristics of survey participants by country and risk group.**

| | | Sex (%) | | Age (years) | | | Education level (%) | | | |
|---|---|---|---|---|---|---|---|---|---|---|
| | | Male | Female | Minimum | Maximum | Average | Up to primary | Secondary | Technical | Higher education |
| **Elderly adults (n = 802)** | Brazil | 28.8 | 71.3 | 62 | 92 | 71.3 | 31.3 | 12.5 | 28.1 | 28.1 |
| | Chile | 21.6 | 78.4 | 65 | 89 | 71.6 | 27.8 | 45.7 | 8.6 | 17.9 |
| | Paraguay | 35.6 | 64.4 | 65 | 91 | 69.8 | 48.1 | 14.4 | 16.9 | 20.6 |
| | Peru | 38.8 | 61.3 | 65 | 95 | 73.3 | 39.4 | 39.4 | 10.0 | 11.3 |
| | Uruguay | 36.9 | 63.1 | 65 | 92 | 73.1 | 47.5 | 20.6 | 11.9 | 20.0 |
| | All | 32.3 | 67.7 | 62 | 95 | 71.8 | 38.8 | 26.6 | 15.1 | 19.6 |
| **Adults with risk factors (n = 808)** | Brazil | 38.1 | 61.9 | 18 | 79 | 49.2 | 22.0 | 14.9 | 35.7 | 27.4 |
| | Chile | 16.3 | 83.8 | 21 | 73 | 45.0 | 19.4 | 33.1 | 13.1 | 34.4 |
| | Paraguay | 25 | 75 | 29 | 80 | 53.3 | 38.8 | 19.4 | 16.9 | 25.0 |
| | Peru | 39.4 | 60.6 | 23 | 96 | 58.0 | 23.1 | 38.1 | 21.3 | 17.5 |
| | Uruguay | 33.8 | 66.3 | 21 | 64 | 49.8 | 31.3 | 24.4 | 23.1 | 21.3 |
| | All | 30.6 | 69.4 | 18 | 96 | 51.0 | 26.9 | 25.9 | 22.2 | 25.1 |
| **Pregnant women (n = 801)** | Brazil | - | 100 | 18 | 42 | 28.2 | 5.6 | 14.4 | 48.8 | 31.3 |
| | Chile | - | 100 | 18 | 41 | 27.8 | 3.8 | 39.4 | 17.5 | 39.4 |
| | Paraguay | - | 100 | 14 | 42 | 27.5 | 8.1 | 25.6 | 18.8 | 47.5 |
| | Peru | - | 100 | 15 | 41 | 27.0 | 3.8 | 46.3 | 27.5 | 22.5 |
| | Uruguay | - | 100 | 16 | 48 | 28.2 | 19.4 | 28.8 | 27.5 | 24.4 |
| | All | - | 100 | 14 | 48 | 27.8 | 8.1 | 30.9 | 28.0 | 33.0 |
| **Mothers of children <6 (n = 800)** | Brazil | - | 100 | 18 | 50 | 31.0 | 4.4 | 8.2 | 47.2 | 40.3 |
| | Chile | - | 100 | 18 | 48 | 30.3 | 4.4 | 38.8 | 22.5 | 34.4 |
| | Paraguay | - | 100 | 16 | 47 | 29.9 | 15.6 | 19.4 | 29.4 | 35.6 |
| | Peru | - | 100 | 16 | 50 | 31.9 | 6.9 | 36.3 | 38.8 | 18.1 |
| | Uruguay | - | 100 | 16 | 49 | 29.3 | 15.0 | 36.3 | 21.3 | 27.5 |
| | All | - | 100 | 16 | 50 | 30.5 | 9.3 | 27.8 | 31.8 | 31.2 |

Table 3 presents the score values for the 3C components together with the statistical results of comparisons across risk groups and countries. All differences are at p < .001 unless otherwise noted. The confidence indicator for adults with risk factors (6.51) is slightly higher than for the rest of the risk groups (6.27 to 6.37). Adults with risk factors and mothers have significantly lower scores for complacency (4.86 and 4.78), meaning they are more complacent–than scores for pregnant women and older adults (4.72 and 4.78). Scores for knowledge of the vaccine are lower for older adults (3.53) followed by adults with risk factors (3.93), with higher and similar scores for the other two groups. Scores for the perception of influenza risks are not significantly different across groups. The convenience score shows minor but significant differences between mothers with a higher score (7.67) in comparison to the other risk groups (7.36 to 7.46).

Self-reported vaccination rates across risk groups were significantly different only for vaccination in the last year. Children and older adults report significantly higher levels of vaccination (56.4 and 51.6%, respectively) than the older adults (45.7%) and pregnant women (48.9%). Vaccination at least once in the lifetime ranged from 60.3 to 64.1% across risk groups.

The analysis of the 3C scores by country shows greater differences than those observed across risk groups. Scores for the confidence indicator are significantly different across all countries, with Chile (7.48) and Uruguay (5.99) at the extremes. Complacency–with less contrasts across countries–shows significant differences between Brazil as least complacent (4.98), Paraguay, Peru and Chile (4.81 to 4.79) and Uruguay (4.45). However, the scores for the sub-components of complacency show marked contrasts across countries. The prejudices subcomponent score groups Paraguay and Peru with least prejudices (5.29 and 5.24), followed by Brazil and Uruguay (4.93 and 5.16) and then by Chile (4.21). With regard to the score for vaccine knowledge, Brazil and Chile rank highest (4.11 and 4.02), followed by Peru, Paraguay and Uruguay (31.4 to 3.81), in that order. The perception of influenza risk scores highest in Chile (5.94), followed by Brazil and Paraguay and, without significant differences, by Peru and Uruguay (5.33 to 5.27).

**Table 3. Comparison across risk groups and countries between the means of individual scores of confidence, complacency, convenience * and the percentage of vaccinated individuals.**

| Indicator | Risk group | | | | Country | | | | | Total |
|---|---|---|---|---|---|---|---|---|---|---|
| | Older adults (n = 802) | Adults with risk factors (n = 808) | Pregnant women (n = 801) | Mothers (n = 800) | Brazil (n = 649) | Chile (n = 642) | Paraguay (n = 640) | Peru (n = 640) | Uruguay (n = 640) | (n = 3,211) |
| Confidence | 6.37 (a) | 6.51 (b) | 6.27 (a) | 6.36 (a) | 5.78 (A) | 7.48 (E) | 6.2 (C) | 6.46 (D) | 5.99 (B) | 6.38 |
| Complacency | 4.72 (a) | 4.86 (c) | 4.65 (a) | 4.78 (b) | 4.98 (C) | 4.72 (B) | 4.81 (B) | 4.79 (B) | 4.45 (A) | 4.75 |
| -Less prejudiced about the vaccine | 4.98 (a) | 5.08 (b) | 4.82 (a) | 4.98 (a) | 5.16 (B) | 4.21 (A) | 5.29 (C) | 5.24 (C) | 4.93 (B) | 4.97 |
| -Knowledge of influenza and vaccine | 3.53 (a) | 3.93 (c) | 3.68 (b) | 3.8 (b) | 4.11 (D) | 4.02 (D) | 3.6 (B) | 3.81 (C) | 3.14 (A) | 3.74 |
| -Perception of risk of influenza | 5.63 (a) | 5.58 (a) | 5.44 (a) | 5.55 (a) | 5.68 (C) | 5.94 (D) | 5.53 (B) | 5.33 (A) | 5.27 (A) | 5.55 |
| Convenience | 7.36 (a) | 7.44 (a) | 7.46 (a) | 7.67 (b) | 7.53 (B) | 8.65 (C) | 6.85 (A) | 6.85 (A) | 7.53 (B) | 7.48 |
| Vaccinated at least once in life course (%) | 61.7 (a) | 60.3 (a) | 63.2 (a) | 64.1 (a) | 78.6 (D) | 87.9 (E) | 51.4 (B) | 58.4 (C) | 35.0 (A) | 6.23 |
| Vaccinated in the last year (%) | 51.6 (b) | 45.7 (a) | 48.9 (a) | 56.4 (b) | 66.3 (C) | 79.1 (D) | 37.5 (B) | 42.5 (B) | 27.5 (A) | 5.06 |

*Scale of 0 to 10 according to more confidence and convenience and less complacency (less prejudice, more knowledge and greater risk perception). In parenthesis, the same letter is assigned to means without statistically significant difference across risk groups or countries at p < .001.

Vaccination status is also highly contrasting across countries, with significant differences in the percentages of individuals vaccinated across all of them and for both status indicators, except for vaccination in the last year which are not significantly different for Paraguay and Peru. Chile shows the highest levels of vaccination in the life-course (87.9%) and in the last year (79.1%), followed by Brazil also for both indicators. Uruguay has the lowest vaccination rates, with 35.0% for vaccination in the life-course and 27.5% for vaccination in the last year.

With regard to the association between the 3C indicator scores and independent sociodemographic variables (Table 4), the score for confidence was significantly associated to older age in the case of pregnant women, to more education in the case of older adults and for the sample as a whole, and with sex only for the sample as a whole (considering only the groups of older adults and adults with risk factors), with male respondents being more confident than female respondents. In the case of complacency, older age was significantly associated with pregnant women ($p < .05$) and to mothers and for the sample as a whole ($p < .01$). (Less) complacency was associated with more education for all risk groups ($p < .01$), and not at all with sex. In the case of convenience, older age was only associated for mothers ($p < .05$) while more education was associated for older adults ($p < .01$), adults with risk factors ($p < .05$) and for the sample as a whole ($p < .01$). Sex was also associated in the case of convenience for adults with risk factors and for the sample as a whole ($p < .01$), with females finding in general vaccine services more convenient.

Multivariate logistic regression analysis shows a significant and positive association between each of the 3C indicators and vaccination status across each risk group ($p < .001$ for all associations), except for the relationship between complacency and vaccination in the last year among older adults ($p < .005$) and among pregnant women which is not significant (Table 5). Across risk groups and considering both vaccination statuses, confidence has the highest odds ratio among older adults (OR: 1.709) and pregnant women (OR: 1.562) while complacency shows the highest odds ratios in the case of adults with risk factors (OR: 1.715) followed by older adults (OR 1.662). Comparing odds ratio for vaccination across the 3Cs,

**Table 4. Association between sociodemographic variables in relation to the 3C indices by risk group, using analysis of variance models.**

| | | Confidence | | Complacency | | Convenience | |
|---|---|---|---|---|---|---|---|
| | | **A** | **B** | **A** | **B** | **A** | **B** |
| **Age** | | | | | | | |
| | OA | NS | - | NS | - | NS | - |
| | ARF | NS | - | NS | - | NS | - |
| | PW | $< .05$ | (+) | $< .05$ | (+) | NS | - |
| | MCh | NS | - | $< .01$ | (+) | $< .05$ | (+) |
| **Education** | | | | | | | |
| | OA | $< .01$ | (+) | $< .01$ | (+) | $< .01$ | (+) |
| | ARF | NS | - | $< .01$ | (+) | $< .05$ | (+) |
| | PW | NS | - | $< .01$ | (+) | NS | - |
| | MCh | NS | - | $< .01$ | (+) | NS | - |
| **Sex** | | | | | | | |
| | OA | NS | - | NS | - | NS | - |
| | ARF | NS | - | NS | - | $< .01$ | M-F |

A. Significance (p-value), NS: Not significant. B. Tendency of association: For Education and Age: (+) positive; (-) negative. For Sex, extreme values are indicated: sex with lower value–sex with higher value. M (Male), F (Female). OA: Older adults; ARF: Adults with risk factors; PW: pregnant women; MCh: mothers of children <6.

**Table 5. Odds ratio of influenza vaccination (at least once in the lifetime and in last year) and vaccine confidence, complacency and convenience, by risk group. All countries.** Binary logistic multivariate regression analyses (enter method to select variables).

| | | OR | CI | p-value | OR | CI | p-value | OR | CI | p-value | |
|---|---|---|---|---|---|---|---|---|---|---|---|
| Older adults | A | 1.635 | 1.429–1.871 | < .001 | 1.662 | 1.372–2.012 | < .001 | 1.247 | 1.112–1.398 | < .001 | |
| | B | 1.709 | 1.501–1.944 | < .001 | 1.268 | 1.076–1.494 | < .005 | 1.220 | 1.095–1.358 | < .001 | |
| Adults with risk factors | A | 1.311 | 1.183–1.452 | < .001 | 1.715 | 1.448–2.031 | < .001 | 1.289 | 1.168–1.422 | < .001 | |
| | B | 1.361 | 1.229–1.507 | < .001 | 1.414 | 1.213–1.648 | < .001 | 1.337 | 1.211–1.475 | < .001 | |
| Pregnant women | A | 1.562 | 1.368–1.784 | < .001 | 1.272 | 1.070–1.513 | < .001 | 1.485 | 1.336–1.650 | < .001 | |
| | B | 1.437 | 1.270–1.625 | < .001 | 1.093 | 0.938–1.273 | NS | 1.400 | 1.267–1.546 | < .001 | |
| Mothers of children | A | 1.299 | 1.150–1.467 | < .001 | 1.382 | 1.168–1.636 | < .001 | 1.381 | 1.239–1.541 | < .001 | |
| | B | 1.313 | 1.170–1.473 | < .001 | 1.346 | 1.152–1.572 | < .001 | 1.322 | 1.192–1.465 | < .001 | |

OR: Odds ratio; CI: Confidence Interval (95%). A: Vaccinated at least once in the lifetime; B: Vaccinated in the last year.

odds ratios are generally lower for convenience, although for pregnant women and mothers these odds are somewhat higher or at the same level than for complacency (OR: 1.040 and 1.033, respectively).

## Discussion

The study is the first to apply a multi-country quantitative approach to the analysis of influenza vaccination hesitancy in South America. The study is limited to non-probabilistic samples of large urban health service user populations and must be interpreted with caution, as health service users may in general find services more accessible and may thus manifest less hesitancy towards vaccines and show also higher vaccination rates than non-users. However, the age and insurance protection in the surveyed population as well as vaccination coverage rates were found to be close to those observed in other more representative sources of information. Most respondents were women, not only given their role in pregnancy and as mothers, but also being over-represented in the groups of older adults and adults with risk factors [7]. It is noteworthy that reported vaccination rates in the last year across risk groups and countries are somewhat lower than those officially reported, particularly for the case of adults with risk factors [8]. This could be due to differences in measurement methods. On the demand side, reporting may be influenced by loss of memory. On the demand side, overreporting can occur given that coverage is estimated on the number of doses applied against the number of doses planned. Planification is particularly difficult for adults with risk factors given limited information on chronic disease prevalence.

The analysis of the 3C indicator levels by risk groups revealed small but significant differences across risk groups, except for influenza risk perception. Adults with risk factors show in general somewhat better indicators across the 3Cs while older adults were always placed in the lowest scores across all indicators, while all other risk groups scored higher than them in the knowledge sub-indicator. Differences across risk groups can support discussion of specific strategies to bolster confidence and convenience and to reduce complacency. In contrast with the 3C levels, we found important differences across the vaccination rates reported by risk groups, with the lowest value for adults with risk factors and the highest for children under six years of age.

Our study found the most significant and contrasting differences in the 3C component indicator levels and vaccination rates across countries. Chile is associated to highest confidence and highest vaccine and influenza knowledge as well as to the highest perception of influenza risks, but also to the highest level of perception of prejudices to the vaccine. Importantly,

respondents from Chile also reported the highest rates of vaccination, coincident with official reports [8]. At the other end, respondents from Uruguay observed the lowest confidence, the most complacency and the least knowledge, together with the lowest reported vaccination rates, also congruent with the official reports. Brazil, Paraguay and Peru rank in the middle of the measured 3C component indicators.

Differences across countries in the 3C component indicators and vaccination rates point to contextual and vaccination program differences. With exceptions, we did not find associations between age and sex with confidence or convenience. Education was associated to complacency for all risk groups but only to confidence for older adults and for the sample as a whole. Education by country in our sample is aligned to the 3C and vaccination rate findings, with respondents in Chile reporting the lowest number with up to primary education while those from Uruguay and Paraguay, the highest.

Our research supports previous findings attesting to the importance of confidence and complacency in hesitancy to influenza vaccination uptake [14, 15]. Reinders and colleagues found confidence as the most prevalent reason for not being vaccinated, specifically "being afraid of vaccination and its effects". They also found greater vaccination rates among persons who most perceived the severity of influenza [16]. We found confidence to be the hesitancy factor most associated to vaccination uptake, particularly among older adults and pregnant women, with complacency being more important among adults with risk factors. A one-point increase in the confidence indicator among older adults would lead to an expected increment of 49% in the probability of being vaccinated at least once; in the case of complacency among adults with risk factors the expected increment would be of 54% and in the case of convenience among minors the increment would be of 32%. These values suggest the importance of addressing influenza vaccination through specific strategies to address each of the 3C indicators.

The COVID-19 pandemic and its devastating effects at the global level calls to prepare vaccination policies and strategies at global, regional and country levels [17]. Our study results point toward valuable lessons to ensure that vaccine confidence is bolstered through effective communication on its expected effectiveness and safety. It is likely COVID-19 has already decreased seasonal influenza vaccine complacency; however, our results suggest that complacency should not take it for granted, and that COVID-19 vaccination when available–should address complacency as a potential barrier to effective coverage.

## Conclusions and recommendations

The analyses carried out on the demand side of influenza vaccination in urban areas of South America enabled the identification of strong benefits from increasing the three 3C components of vaccine hesitancy, particularly confidence and complacency. The study suggests diverse opportunities and pathways are open to strengthening the 3C components, which show greater variation across countries than across risk groups. The higher values identified for vaccine convenience suggest this component has been more successful in the implementation of influenza vaccination campaigns, although opportunities still exist for improvement. Strategies to bolster confidence and to reduce complacency should be supported on the strengths of vaccine convenience.

It is recommended to investigate the contextual and vaccination program factors that are affecting the perception of confidence and complacency and vaccination rates across the countries studied and in relation to each of the risk groups. Information and communication campaigns, normative messaging and performance incentives can be strengthened to focus on those aspects that affect confidence and complacency. With the advent of COVID-19 lessons

learnt with influenza vaccination can play a useful role to identify strategies to ensure confidence and reduce complacency and to support vaccination on the existing influenza program platform.

## Supporting information

**S1 Questionnaire.**
(PDF)

**S1 File.**
(PDF)

## Author Contributions

**Conceptualization:** Emilio Gutiérrez-Calderón, Blanca Estela Pelcastre-Villafuerte, Elsa Sarti, Esteban Puentes-Rosas.

**Formal analysis:** Miguel Ángel González-Block, Emilio Gutiérrez-Calderón, Blanca Estela Pelcastre-Villafuerte.

**Investigation:** Miguel Ángel González-Block, Juan Arroyo-Laguna, Yamila Comes, Pedro Crocco, Andréa Fachel-Leal, Laura Noboa, Daniela Riva-Knauth, Berenice Rodríguez-Zea, Mónica Ruoti, Elsa Sarti, Esteban Puentes-Rosas.

**Methodology:** Miguel Ángel González-Block, Juan Arroyo-Laguna, Yamila Comes, Pedro Crocco, Andréa Fachel-Leal, Laura Noboa, Daniela Riva-Knauth, Berenice Rodríguez-Zea, Mónica Ruoti.

**Writing – original draft:** Miguel Ángel González-Block.

**Writing – review & editing:** Emilio Gutiérrez-Calderón, Blanca Estela Pelcastre-Villafuerte, Juan Arroyo-Laguna, Yamila Comes, Pedro Crocco, Andréa Fachel-Leal, Laura Noboa, Daniela Riva-Knauth, Berenice Rodríguez-Zea, Mónica Ruoti, Elsa Sarti, Esteban Puentes-Rosas.

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
