## [Decision Letter · Decision Letter 0]

4 Sep 2020

PONE-D-20-15450

Influenza vaccination hesitancy in five countries of South America. Confidence, complacency and convenience as determinants of immunization rates

PLOS ONE

Dear Dr. Gonzalez-Block,

Thank you for submitting your manuscript to PLOS ONE. After careful consideration, we feel that it has merit but does not fully meet PLOS ONE’s publication criteria as it currently stands. Therefore, we invite you to submit a revised version of the manuscript that addresses the points raised during the review process.

Both reviewers have provided very positive comments about your paper but have also stated that there are critical issues with the approach taken to the analysis that need to be teased out. Can I please encourage you to carefully go through these issues and update your paper accordingly

We look forward to receiving your revised manuscript.

Kind regards,

Holly Seale

Academic Editor

PLOS ONE

2. Please address the following:

- Please ensure you have thoroughly discussed any potential limitations of this study within the Discussion section.

- Please include additional information regarding the survey or questionnaire used in the study and ensure that you have provided sufficient details that others could replicate the analyses. For instance, if you developed a questionnaire as part of this study and it is not under a copyright more restrictive than CC-BY, please include a copy, in both the original language and English, as Supporting Information. In addition, please include any details concerning the validation of this tool, i.e. any pre-testing that took place.

3. Thank you for including your competing interests statement; "This project was financially supported by Sanofi Pasteur. Elsa Sarti and Esteban Puentes-Rosas are employees of Sanofi Pasteur. All other authors have no relevant conflicts of interest to report."

4. Thank you for including your ethics statement:  "Brazil, Comissao Nacional de Ética em Pesquisa, 05215918.6.0000.5347.

Chile: Comité de Ética de Investigación en Seres, Universidad de Chile, Facultad de

Medicina, 191-2018.

Paraguay: Comité de ética en Investigación, Laboratorio Central de Salud Pública,

106/2019.

Peru: Comité de Ética de Investigación Prisma, CE1651.18.

Uruguay: Comité de Ética en Investigación, Instituto Nacional de Salud Pública, 1580.".   

Please amend your current ethics statement to confirm that your named institutional review board or ethics committee specifically approved this study.

Reviewers' comments:

Reviewer's Responses to Questions

**Comments to the Author**

1. Is the manuscript technically sound, and do the data support the conclusions?

Reviewer #1: Yes

Reviewer #2: Yes

2. Has the statistical analysis been performed appropriately and rigorously? 

Reviewer #1: No

Reviewer #2: I Don't Know

3. Have the authors made all data underlying the findings in their manuscript fully available?

Reviewer #1: No

Reviewer #2: No

4. Is the manuscript presented in an intelligible fashion and written in standard English?

Reviewer #1: Yes

Reviewer #2: Yes

5. Review Comments to the Author

Reviewer #1: Overview:

The paper reports the rates of vaccine hesitancy in respect to the adult influenza vaccine across five countries within South America. The confidence, complacency and convenience framework is use to assess different aspects of vaccine hesitancy. These factors are compared across risk groups, countries and to self-reported influenza vaccine uptake.

General comments:

The paper is well written and the topic is novel and importance to our understanding of global vaccine hesitancy. The data set clearly contains some important finding, however, the way in which you report the 3c factors and some of the statistical tests are confusing and would benefit from reworking.

In this section I have outlined the 2 major issues from my perspective. Below this I have a few specific and minor comments.

1. Reporting of the 3c subscales:

- As I understand it you have added the items of each of the 3c sub scales and then turned them into a number between 1 and 100 so that they are comparable. Then you have reported these as a percentage. I concerned that this makes your findings confusing, as when you report percentages it makes your finding sound like each component is a binary and the percentage is referring to how many people are e.g. complacent/not complacent (i.e. not a scale variable)

- I suggest change these to a number between 1 and 10. This would make reporting clearer and distinguish these from where you report percentages of categorical variable (e.g. when you report vaccine uptake rates). If this change is made the analysis would need to be re-run, however, the conclusions would obviously remain the same.

- Table 1 confuses the issue further. For the “Confidence in the vaccine” indicator you write “Resulting sum is rescaled to the interval (0,1) and expressed as %”. If I was looking at this out of context, I would assume that you had split the sample into high and low confidence and were reporting the percentage of those in one of the categories.

2. Comparison ANOVAs and Tables 3:

- The use of letters to signify significance makes this table difficult to understand. It also means that the reader cannot see the level of significance or the effect size. My advice would be to check other papers in Plos One and see how they report multiple ANOVAs like this.

- Type 1 errors: With the number of comparison ANOVA tests that you have conducted I am concerned with the possibility of false positives in your conclusions. At minimum a note of caution should be included in the discussion, however, I personally would reduce the alpha level throughout. This would of course change the conclusions of your analysis, especially for those results that are borderline significant, however, it would remove any spurious results.

Minor comments:

Line 105: Cross-section (rather than transversal) would be a more accessible, and the standard, term to use for this type of study.

Line 119 -120: Please direct the reader to the section with the listed ethics details here.

Line 146-147: Talking in terms of a higher and lower values and not “100 means less complacently” e.g. a higher value on complacence variable indicates a lower level of complacence.

Line 169-172: As per the clarity of the 3c’s comment these would then be: “the highest normalised score for convenience (7.48, out of a possible 10)” etc.

Line 173 - 182: Each comparison in this section should be accompanied with the reporting of a statistical test to demonstrate significant differences. Or point towards a table that reports all comparisons.

Line 183 -185: This sentence would benefit from rephrasing. Something like “…with children and older adults reporting significantly higher levels of vaccination than the other two risk groups”, then point to a table reporting the means and statistical test.

Line 202 -213: If these are correlations, please report the correlation coefficients for each. Either in the text or on the table.

Table 1:

- Change “Points added gradually” to “The scale was additive”

- In “Complacency A. Influenza risk” construction you have the word “Idem.”, it is unclear what this means.

Table 5:

- Report all p values without first “0”, i.e. “<.01”

- A p< .001 level of detail would be useful here.

Supplemental materials:

- Some parts of the questionnaire in the supplemental materials still require translating.

- Data set should be made fully available

Reviewer #2: This manuscript will make an important contribution to the larger literature on influenza vaccine hestiancy, particularly as it covers immunization in an understudied region. However, I think a revision is necessary to strengthen the connections between the vaccine hestiancy framework, particularly the 3 C's, and the study's research questions and conclusions.

Introduction:

I would have liked to see more explicit research questions. There are a lot of different factors at play here -- multiple risk groups, many different countries, several different theoretical constructs, etc. Outlining research questions may help the reader focus on the significance of this work and keep both the results and discussion sections more focused.

Given the title of the paper, I was expecting more focus on the 3 C's. I find this overview of the constructs to be a bit superficial and lacking in support. Please incorporate more of the existing literature on vaccine hesitancy. Given that the conceptualizations of constructs at this stage is carried through to the development of survey measures and in analysis, this has significant bearing on the meaning of the results.

Although a very different population, a recent study by Quinn et al. 2019 may be helpful. Quinn, S. C., Jamison, A. M., An, J., Hancock, G. R., & Freimuth, V. S. (2019). Measuring vaccine hesitancy, confidence, trust and flu vaccine uptake: Results of a national survey of White and African American adults. Vaccine, 37(9), 1168-1173.

Ln 68: What does "mandated" mean in this context? I associate this with mandatory or required vaccination, yet the following sentences seem to suggest this is not the case.

Ln 73: What would an ideal vaccination rate be? I was surprised to see rates this high.

Methods:

As alluded to earlier, more information is needed on item development and any reliability/validity tests that were performed. Also more information on the choice to combine all respondents into a single "high-risk" sample may be needed. What differences exist between risk groups and between countries? I need to be convinced that the measures are actually measuring what the authors claim they are.

Ln 121: How does this sampling approach introduce bias? In what ways did the resulting sample differ from a more resprsentative sample?

Ln 133: Since the 3 C's are so central to the arguements of this paper, it is worth describing the development of measures in greater detail in the body of the text, not just in a table. Specifically, what each indicator was designed to measure and then how well it measures it.

Results:

The indicators scores need to be contextualized to be meaningful. What does a convenience indicator of 74.8% mean in this context? I'm not sure what I'm supposed to take away from these results. I'm also not sure what is a "good" or "acceptable" level for any of these indicators and which are "bad" or "unacceptable" levels. Starting with how these scores correlate to vaccine behavior may be more helpful.

ln 183: Why are vaccination rates so much lower than those described in the introduction? Any ideas?

Ln 200: The differences in vaccination rates between Chile and Uruguay are pretty extreme. How does this impact the dataset? Did the 3 C's function the same across different countries?

Discussion:

With so many results to touch upon, the discussion section feels a little unfocused. Again, tying the paper more closely to the vaccine hestiancy framework and following a clear set of research questions may help tighten the focus of the paper.

Some of these statments need to be further explained in much more detail. For instance, Ln 238-239. What does it mean to suggest that convenience is better than confidence and both are better than complacency?

Ln 301 this paragraph feels more like a literature review than a part of the discussion. How is it tied to the study's findings?

6. PLOS authors have the option to publish the peer review history of their article (what does this mean?). If published, this will include your full peer review and any attached files.

Reviewer #1: **Yes: **Dr Richard M Clarke

Reviewer #2: No

---

## [Author Response · Author response to Decision Letter 0]

31 Oct 2020

Response to peer review PONE-D-20-15450

General comments Reviewer #1 The paper reports the rates of vaccine hesitancy in respect to the adult influenza vaccine across five countries within South America. The confidence, complacency and convenience framework is use to assess different aspects of vaccine hesitancy. These factors are compared across risk groups, countries and to self-reported influenza vaccine uptake. We appreciate these comments.

 Reviewer #2 This manuscript will make an important contribution to the larger literature on influenza vaccine hestiancy, particularly as it covers immunization in an understudied region. However, I think a revision is necessary to strengthen the connections between the vaccine hestiancy framework, particularly the 3 C's, and the study's research questions and conclusions. We appreciate these comments.

 Reviewer #1 The paper is well written and the topic is novel and importance to our understanding of global vaccine hesitancy. The data set clearly contains some important finding, however, the way in which you report the 3c factors and some of the statistical tests are confusing and would benefit from reworking. We appreciate these comments.

Introduction Reviewer #2 I would have liked to see more explicit research questions. There are a lot of different factors at play here -- multiple risk groups, many different countries, several different theoretical constructs, etc. Outlining research questions may help the reader focus on the significance of this work and keep both the results and discussion sections more focused. We appreciate the comments and suggestions to focus on the research questions. We now propose three research questions and clarify the research objective.

 Given the title of the paper, I was expecting more focus on the 3 C's. I find this overview of the constructs to be a bit superficial and lacking in support. Please incorporate more of the existing literature on vaccine hesitancy. Given that the conceptualizations of constructs at this stage is carried through to the development of survey measures and in analysis, this has significant bearing on the meaning of the results. Although a very different population, a recent study by Quinn et al. 2019 may be helpful. Quinn, S. C., Jamison, A. M., An, J., Hancock, G. R., & Freimuth, V. S. (2019). Measuring vaccine hesitancy, confidence, trust and flu vaccine uptake: Results of a national survey of White and African American adults. Vaccine, 37(9), 1168-1173. We are thankful for this comment. We now provide a more in-depth overview of the 3C model based on the quantitative research reported in the literature for influenza vaccination hesitancy.

 Ln 68: What does "mandated" mean in this context? I associate this with mandatory or required vaccination, yet the following sentences seem to suggest this is not the case. We substitued "mandated" with "recommended and publicily financed for"

 Ln 73: What would an ideal vaccination rate be? I was surprised to see rates this high. We now cite WHO recommendations of reachibng desirable vaccination rates of 90% for 2020.

Methods Reviewer #1 1. Reporting of the 3c subscales: 

 - As I understand it you have added the items of each of the 3c sub scales and then turned them into a number between 1 and 100 so that they are comparable. Then you have reported these as a percentage. I concerned that this makes your findings confusing, as when you report percentages it makes your finding sound like each component is a binary and the percentage is referring to how many people are e.g. complacent/not complacent (i.e. not a scale variable)

 I suggest change these to a number between 1 and 10. This would make reporting clearer and distinguish these from where you report percentages of categorical variable (e.g. when you report vaccine uptake rates). If this change is made the analysis would need to be re-run, however, the conclusions would obviously remain the same. 

We have rescaled the variables as suggested to report our results more clearly.

 Table 1 confuses the issue further. For the “Confidence in the vaccine” indicator you write “Resulting sum is rescaled to the interval (0,1) and expressed as %”. If I was looking at this out of context, I would assume that you had split the sample into high and low confidence and were reporting the percentage of those in one of the categories. Text in the table is now restated in terms of the new categorization.

 2. Comparison ANOVAs and Table 3: 

 The use of letters to signify significance makes this table difficult to understand. It also means that the reader cannot see the level of significance or the effect size. My advice would be to check other papers in Plos One and see how they report multiple ANOVAs like this. We are maintaining the use of letters to represent significance as this device allows us to compare significant differences between pairs of countries. However, we now specify the alpha level in the table footer to the level <.001

 Type 1 errors: With the number of comparison ANOVA tests that you have conducted I am concerned with the possibility of false positives in your conclusions. At minimum a note of caution should be included in the discussion, however, I personally would reduce the alpha level throughout. This would of course change the conclusions of your analysis, especially for those results that are borderline significant, however, it would remove any spurious results. We have reduced the alpha level to <.001. Most results were significant at this level, except when noted.

 Reviewer #2 As alluded to earlier, more information is needed on item development and any reliability/validity tests that were performed. Ln 133: Since the 3 C's are so central to the arguements of this paper, it is worth describing the development of measures in greater detail in the body of the text, not just in a table. Specifically, what each indicator was designed to measure and then how well it measures it. We now report on the procedure used for the development of indicators and cite the relevant references.

 Also more information on the choice to combine all respondents into a single "high-risk" sample may be needed. What differences exist between risk groups and between countries? I need to be convinced that the measures are actually measuring what the authors claim they are. Reflecting on the pertinence of combining all respondents ino a single group, we coma to the conclusion that it is not meaningful and have therefore deleted in tables and text reference to these aggregate values. We focus now the analysis on differences for each indicator across risk groups and countries.

 Ln 121: How does this sampling approach introduce bias? In what ways did the resulting sample differ from a more resprsentative sample? We now discuss in the limitation of the study the possible biases of our sampling strategy with respect to service utilization.

Results Reviewer #2 The indicators scores need to be contextualized to be meaningful. What does a convenience indicator of 74.8% mean in this context? I'm not sure what I'm supposed to take away from these results. I'm also not sure what is a "good" or "acceptable" level for any of these indicators and which are "bad" or "unacceptable" levels. Starting with how these scores correlate to vaccine behavior may be more helpful. We have revised the interpretation of the 3C indicators to avoid comparison across them, and limit comparison within each indicator across risk groups and countries. However, we consider it reasonable to compare indicators across each other with respect to their association with vaccination status.

 ln 183: Why are vaccination rates so much lower than those described in the introduction? Any ideas? We take up in the Discussion the differences between reported national vaccionation rates and our results. 

 Ln 200: The differences in vaccination rates between Chile and Uruguay are pretty extreme. How does this impact the dataset? Did the 3 C's function the same across different countries? As stated in the Introduction, countries were chosen across the continuum of vaccionation coverage. We discuss differenvrd in the 3C and reported vaccination rates across countries in lines 622 to 636.

Discussion Reviewer #2 With so many results to touch upon, the discussion section feels a little unfocused. Again, tying the paper more closely to the vaccine hestiancy framework and following a clear set of research questions may help tighten the focus of the paper. We have formulated three research questions and refocused the discussion accordingly. 

 Some of these statments need to be further explained in much more detail. For instance, Ln 238-239. What does it mean to suggest that convenience is better than confidence and both are better than complacency? Following from our response to reviewer 2' s comment for the Results section above, we now avoid comparing across the 3C indicators, except when discussing their association wih vaccination status. We have therefore removed the disciussion of comparisons across indicators.

 Ln 301 this paragraph feels more like a literature review than a part of the discussion. How is it tied to the study's findings? We now integrate the two references in this paragrpah as similar evidence that is similar to our study on the importance of confidence and complacency.

Minor comments: Reviewer #1 Line 105: Cross-section (rather than transversal) would be a more accessible, and the standard, term to use for this type of study. We have changed the wording

 Line 119 -120: Please direct the reader to the section with the listed ethics details here. We have directed the reader to the ethics details as suggested

 Line 146-147: Talking in terms of a higher and lower values and not “100 means less complacently” e.g. a higher value on complacence variable indicates a lower level of complacence. We have modified the wording accordingly

 Line 169-172: As per the clarity of the 3c’s comment these would then be: “the highest normalised score for convenience (7.48, out of a possible 10)” etc. We have modified the wording accordingly

 Line 173 - 182: Each comparison in this section should be accompanied with the reporting of a statistical test to demonstrate significant differences. Or point towards a table that reports all comparisons. We are now stating that results are at the p<.001 unless otherwise stated. We are also pointing to table 3 for further details. 

 Line 183 -185: This sentence would benefit from rephrasing. Something like “…with children and older adults reporting significantly higher levels of vaccination than the other two risk groups”, then point to a table reporting the means and statistical test. We have modified the wording following the suggestion.

 Line 202 -213: If these are correlations, please report the correlation coefficients for each. Either in the text or on the table. No, these are not correlations. The analysis is based on analysis of variance models.

 Table 1: 

 Change “Points added gradually” to “The scale was additive” We have modified the wording

 In “Complacency A. Influenza risk” construction you have the word “Idem.”, it is unclear what this means. We have modifided the wording, now stating "Same as above".

 Table 5: 

 Report all p values without first “0”, i.e. “<.01” We have modified deleting the 0.

 A p< .001 level of detail would be useful here. We are now reporting at the <.001 level.

 Supplemental materials: 

 Some parts of the questionnaire in the supplemental materials still require translating. We have completed the translation

 Data set should be made fully available We are now making the dataset available.

---

## [Decision Letter · Decision Letter 1]

27 Nov 2020

Influenza vaccination hesitancy in five countries of South America. Confidence, complacency and convenience as determinants of immunization rates

PONE-D-20-15450R1

Dear Dr. Gonzalez-Block,

We’re pleased to inform you that your manuscript has been judged scientifically suitable for publication and will be formally accepted for publication once it meets all outstanding technical requirements.

Kind regards,

Holly Seale

Academic Editor

PLOS ONE

Additional Editor Comments (optional):

Reviewers' comments:

Reviewer's Responses to Questions

**Comments to the Author**

1. If the authors have adequately addressed your comments raised in a previous round of review and you feel that this manuscript is now acceptable for publication, you may indicate that here to bypass the “Comments to the Author” section, enter your conflict of interest statement in the “Confidential to Editor” section, and submit your "Accept" recommendation.

Reviewer #1: All comments have been addressed

2. Is the manuscript technically sound, and do the data support the conclusions?

Reviewer #1: Yes

3. Has the statistical analysis been performed appropriately and rigorously? 

Reviewer #1: Yes

4. Have the authors made all data underlying the findings in their manuscript fully available?

Reviewer #1: Yes

5. Is the manuscript presented in an intelligible fashion and written in standard English?

Reviewer #1: Yes

6. Review Comments to the Author

Reviewer #1: All my previous comments appear to have been addressed and I am happy for to recommend the article for publication.

7. PLOS authors have the option to publish the peer review history of their article (what does this mean?). If published, this will include your full peer review and any attached files.

Reviewer #1: **Yes: **Dr Richard M Clarke

---

## [Editor Report · Acceptance letter]

2 Dec 2020

PONE-D-20-15450R1 

Influenza vaccination hesitancy in five countries of South America. Confidence, complacency and convenience as determinants of immunization rates 

Dear Dr. González-Block:

I'm pleased to inform you that your manuscript has been deemed suitable for publication in PLOS ONE. Congratulations! Your manuscript is now with our production department. 

Kind regards, 

on behalf of

Dr. Holly Seale 

Academic Editor

PLOS ONE